# The Underlying Biology and Therapeutic Vulnerabilities of Leptomeningeal Metastases in Adult Solid Cancers

**DOI:** 10.3390/cancers13040732

**Published:** 2021-02-10

**Authors:** Matthew Dankner, Stephanie Lam, Theresa Degenhard, Livia Garzia, Marie-Christine Guiot, Kevin Petrecca, Peter M. Siegel

**Affiliations:** 1Goodman Cancer Research Centre, Faculty of Medicine, McGill University, Montreal, QC H3A 1A3, Canada; matthew.dankner@mail.mcgill.ca (M.D.); marie.christine.guiot@mcgill.ca (M.-C.G.); 2Department of Diagnostic Radiology, Faculty of Medicine, Research Institute of the McGill University Health Centre, Montreal, QC H3T 1E2, Canada; stephanie.lam@mcgill.ca; 3Research Institute of the McGill University Health Centre, Montreal, QC H4A 3J1, Canada; Livia.garzia@mcgill.ca; 4Department of Neurology and Neurosurgery, Montreal Neurological Institute-Hospital, Goodman Cancer Research Centre, McGill University, Montreal, QC H3A 2B4, Canada; theresa.degenhard@mail.mcgill.ca (T.D.); kevin.petrecca@mcgill.ca (K.P.); 5Department of Neurology and Neurosurgery, Montreal Neurological Institute-Hospital, McGill University, Montreal, QC H3A 2B4, Canada; 6Department of Pathology, McGill University, Montreal, QC H3A 1A3, Canada; 7Department of Biochemistry, McGill University, Montreal, QC H3A 1A3, Canada; 8Department of Anatomy & Cell Biology, McGill University, Montreal, QC H3A 1A3, Canada; 9Department of Oncology, McGill University, Montreal, QC H3A 1A3, Canada

**Keywords:** brain metastasis, leptomeningeal metastases, subarachnoid space, breast cancer, lung cancer, melanoma

## Abstract

**Simple Summary:**

Leptomeningeal metastases occur when cancer cells reach the fluid-filled space that surrounds the brain and spinal cord. This type of brain lesion is most prevalent in patients with lung cancer, breast cancer and melanoma. While the clinical characteristics of leptomeningeal metastases have been well described, fundamental research revealing insights into how these lesions develop and grow has only just begun. This review describes the clinical and basic science literature surrounding leptomeningeal metastases from adult solid tumors, proposing novel ways the clinical and research communities can work together to make important advances for patients suffering from this devastating complication of advanced cancer.

**Abstract:**

Metastasis to the central nervous system occurs in approximately 20% of patients with advanced solid cancers such as lung cancer, breast cancer, and melanoma. While central nervous system metastases most commonly form in the brain parenchyma, metastatic cancer cells may also reside in the subarachnoid space surrounding the brain and spinal cord to form tumors called leptomeningeal metastases. Leptomeningeal metastasis involves cancer cells that reach the subarachnoid space and proliferate in the cerebrospinal fluid compartment within the leptomeninges, a sequela associated with a myriad of symptoms and poor prognosis. Cancer cells exposed to cerebrospinal fluid in the leptomeninges must contend with a unique microenvironment from those that establish within the brain or other organs. Leptomeningeal lesions provide a formidable clinical challenge due to their often-diffuse infiltration within the subarachnoid space. The molecular mechanisms that promote the establishment of leptomeningeal metastases have begun to be elucidated, demonstrating that it is a biological entity distinct from parenchymal brain metastases and is associated with specific molecular drivers. In this review, we outline the current state of knowledge pertaining to the diagnosis, treatment, and molecular underpinnings of leptomeningeal metastasis.

## 1. Introduction

Leptomeningeal metastasis (LM), also known as leptomeningeal carcinomatosis or neoplastic meningitis, is a debilitating condition associated with metastatic solid tumors. LM involves cancer cells reaching the subarachnoid space, surviving in the cerebrospinal fluid (CSF) and frequently adhering to the leptomeninges [1]. LM is most commonly reported in solid tumors that include breast cancer, lung cancer, and melanoma. Approximately 4–15% of all the adult patients with solid tumors develop LM [2,3,4], 10% of whom present with LM as their first manifestation of metastatic disease [5]. It is estimated that 30–75% of LM patients have concomitant parenchymal brain metastases [6,7,8,9].

LM is associated with a poor quality of life. Patients often experience symptoms such as headaches, nausea/vomiting, neck/back pain, cranial nerve palsies, and gait difficulties [2]. Patients diagnosed with LM survive an average of 2–6 months [2,9,10,11,12]. Recent studies have clarified that LM is indeed a distinct biological entity from parenchymal brain metastases, indicating that significant pre-clinical and clinical research efforts must be undertaken to better understand and treat LM [13].

There are a number of risk factors that are associated with the development of LM. Surgical resection of parenchymal brain metastases, particularly those in the posterior fossa [14,15], which are accessed via intraventricular entry [16] and/or resected in a piecemeal fashion [14,15,17,18], is associated with the development of LM. In breast cancer, lobular histology, HER2 positive, and triple negative subtypes are associated with a higher likelihood of developing LM [19,20]. In non-small cell lung cancer, EGFR mutations and ALK translocations have been reported to predict an increased likelihood of LM [21,22]. In recent years, the incidence of solid tumor CNS metastases has been rapidly increasing [23]. This trend may be explained by a combination of factors, including improvements to the treatment of extracranial cancers and enhanced imaging modalities with greater detection sensitivity [5]. Once established, CNS metastases are less responsive to chemotherapies due to the impaired drug delivery and rapid efflux imposed by the blood-brain-, blood-CSF- and blood-meningeal barriers [24], and molecular mediators in the brain and LM microenvironments that may dampen the treatment response [25]. Together, it is clear that LM is an emerging issue in cancer care that requires robust clinical and biological investigation.

## 2. Anatomy and Function of the Leptomeninges

The brain and spinal cord are surrounded by tissues called meninges (Figure 1) [26]. Under the skull, the first meningeal layer encountered is a fibrous structure called the pachymeninges, or dura mater. The dura mater contains lymphatic vessels and fenestrated blood vessels with no tight junctions, leaving them open to peripheral circulation [27,28]. Between the two layers of dura mater, the periosteal dura attached to the skull and the meningeal dura adhered to the leptomeninges, are the dural venous sinuses that drain blood and CSF from the brain to the venous circulation. The leptomeninges lie beneath the meningeal dura and consist of the arachnoid and pia mater. The arachnoid mater is an avascular and thin membrane that is contiguous with the meningeal dura and contains epithelial cells with tight junctions [29]. The arachnoid and pia mater is separated by the subarachnoid space, which is filled with CSF. The pia separates the subarachnoid space from the brain but has no tight junctions, allowing for the exchange of fluid from the brain parenchyma and the subarachnoid space through perivascular spaces around blood vessels in the brain [30,31]. The glia limitans, a thin barrier of astrocytic foot processes, is found at the interface of the pia and the brain parenchyma and acts as a physical and immunological barrier that regulates the movement of molecules and cells between the brain and the subarachnoid space. The leptomeninges is believed to be continuous along the entire CNS with few exceptions, such as the circumventricular organs that include the posterior pituitary, pineal gland, subfornical organ, the area postrema, the median eminence, and the vascular organ of the lamina terminalis. These organs are neural structures located near the third and fourth ventricles that are not fully covered by the blood-brain barrier and are therefore, in direct contact with the systemic circulation [32]. Other spaces in the leptomeninges, called the olfactory bulb hole, peripituitary slit and peripineal slit, have been identified in rodents but have yet to be observed in humans [26].

The ventricles within the brain are CSF-filled cavities that contain the choroid plexus, a specialized epithelium that filters arterial blood plasma into clear, protein-poor CSF [33]. The choroid plexus consists of an epithelial cell layer surrounding capillaries [34]. CSF is made by the filtration of plasma from fenestrated capillaries into the interstitial space of the choroid plexus. Subsequently, the CSF is actively transported through the tight junction-containing choroid plexus epithelium into the ventricles by establishing an osmotic gradient of sodium, potassium, and chloride [35]. The resulting CSF bathes the brain and spinal cord and is contained within the subarachnoid space. CSF has been suggested to be recycled back into the circulation through a combination of the venous sinus system, lymphatic vessels, and exchange with the perivascular spaces surrounding the brain’s blood vessels [28,31,36].

The normal composition of CSF is acellular and low in mitogens and glucose [13]. In disease settings, the composition of CSF can change substantially, in part due to the extravasation of immune cells through the choroid plexus into the CSF and conditioning from cancer cells in the context of LM. CSF from patients with LM reveals factors secreted by tumor cells or inflammatory cells recruited to the CSF [13], including vascular endothelial growth factor (VEGF) [37]. The CSF is a unique and particularly harsh microenvironment with low levels of oxygen, glucose, and mitogens [38], suggesting that the formation of LM, such as parenchymal brain metastases [39], is established through a stringent selection imposed by the unique conditions within this microenvironment [13].

## 3. Defining LM by the Route of Entry

Cancer cells have been proposed to reach the leptomeninges through a number of routes including: (1) The venous circulation via Batson’s venous plexus, a network of veins that connect the vertebral column with the peripheral circulation, (2) the arterial circulation via the choroid plexus, (3) invasion of spinal or cranial nerves through the bone, and (4) invasion across the glia limitans from the brain parenchyma [13,40,41,42]. Each of these routes has been suggested in the literature based on limited evidence and may not apply to many or most cases of LM in solid tumor patients.

Alternate routes have become increasingly plausible in recent years due to an improved understanding of fluid dynamics in the CNS. Given that a large proportion of patients with LM have coexisting parenchymal brain metastases [43], it is evident that a subset of LM can form secondary to the establishment of parenchymal brain metastases. The converse may also be true, where leptomeningeal deposits can be observed invading into the brain parenchyma (Figure 2). Metastatic cancer cells have been shown to cycle between the brain parenchyma and leptomeningeal compartments in rodent models [44], with cold-inducible RNA-binding protein (CIRBP) and cyclooxygenase-2 representing candidate genes implicated in this process [44,45]. It is possible that shuttling between the brain and leptomeningeal anatomical compartments is uniquely driven by invasion across the glia limitans, however, other possibilities remain. Cancer cells are frequently observed in the perivascular spaces surrounding blood vessels within the brain and are known to reach the CNS from the periphery by first extravasating out of vessels into this space [46]. These perivascular spaces are in direct contact with the pia mater, which lacks tight-junctions, and are enlarged in proximity to frequent sites of LM establishment at the skull base and cerebellum [30,31]. Therefore, invasion of metastatic cells that are situated within the perivascular spaces surrounding brain blood vessels across the pia and into the subarachnoid space represents an enticing possibility.

The recently discovered lymphatic vessels within dural folds located at the skull base represent another opportunity for cancer cells to reach the leptomeninges [47]. Lymphatic vessels are known conduits for metastasis [48], and their interaction with CSF and the leptomeninges creates an opportunity for cancer cells to extravasate from lymphatic vessels into the subarachnoid space. Moreover, blood vessels associated with the dura mater, which are part of the systemic circulation, are fenestrated in nature, making access of metastatic cells to this anatomical location more likely [28]. Invasion of pachymeningeal deposits into the leptomeninges is another theoretical route of entry. In addition to the aforementioned anatomical routes of LM extravasation, iatrogenic seeding of the leptomeninges following surgical resection of parenchymal brain metastases comprises a subset of LM cases [49].

Understanding the differences between these different routes of LM seeding may have important implications for prognosis and management. It is conceivable that distinct leptomeningeal cancer cell niches may provide differing degrees of protection from systemic therapies. Due to the multiple independent routes that cancer cells use to colonize the leptomeninges, approaches to treat LM may be best focused on the identification of factors that promote cell survival and proliferation within the subarachnoid space rather than trying to target specific cancer cell entry points.

## 4. Diagnosis of LM

The detection and diagnosis of LM is notoriously difficult. LM is diagnosed by combining neurological evaluation, leptomeningeal enhancement on magnetic resonance imaging (MRI) or computed tomography (CT), and/or the identification of tumor cells within the CSF. Recently, the EANO-ESMO clinical practice guidelines for diagnosis, treatment, and follow-up of patients with LM from solid cancers has emerged on the basis of an expert opinion and consensus due to the lack of high-quality clinical trial data [50,51]. This system defines the diagnosis of LM as (1) confirmed (type I) with positive CSF cytology, or (2) probable/possible (type II) with a combination of typical MRI features and clinical signs [50]. This categorization is further refined by an imaging pattern of LM lesions as linear (subtype A), nodular (subtype B), both (subtype C) or neither (subtype D).

### 4.1. Neurological Evaluation

LM is suspected as part of a differential diagnosis when patients with solid cancers present with common symptoms such as cranial nerve deficits, seizures, headaches, and back pain [52]. However, these symptoms are difficult to distinguish from those associated with parenchymal brain metastases or other conditions, so more focused strategies are required to render a LM diagnosis as probable or confirmed.

### 4.2. CSF Cytology

While positive CSF tumor cell cytology is the most specific methodology to confirm the diagnosis and is considered the gold standard for LM diagnosis, this test has low sensitivity [51]. Patients with positive CSF cytology (type 1 LM) have been reported to demonstrate an improved response with intrathecal compared to systemic therapy in retrospective studies [50].

As a result of the challenges associated with diagnosing LM, the research community has been actively engaged in developing improved diagnostic tests to more accurately detect LM. Several groups have developed flow cytometry-based approaches for the detection of breast cancer LM, capturing cells positive for tumor cell specific markers from CSF [53,54,55]. Others have applied rare cell capture technologies to standard CSF cytology approaches to improve the detection sensitivity of tumor cells [56,57,58,59]. Approaches examining the genome, transcriptome, proteome, or miRNAs found in CSF may also hold future promise in diagnosing LM [37,60,61,62]. While these liquid biopsy technologies from CSF are promising in their potential to improve the diagnosis of LM, they have not yet been adopted in diagnostic guidelines or achieved widespread implementation in clinical practice. The gold standard for LM diagnosis remains CSF cytology, and it is yet to be determined whether a superior diagnostic approach, with high sensitivity and specificity, can be achieved with either serum or CSF liquid biopsies [63].

### 4.3. Imaging-Based Approaches

Cerebrospinal MRI is the standard method currently used to visualize leptomeningeal lesions [51]. However, diagnosing LM by MRI remains imperfect due to inter-observer variability, the heterogeneous appearance of leptomeningeal lesions by neuroimaging, and other pathologies that can give rise to contrast enhancing lesions in the leptomeninges [64]. Such conditions may include meningitis, encephalitis, or primary brain tumors, including invasive meningiomas and diffuse leptomeningeal glioneuronal tumors.

LM can display a diverse range of features by MRI. The enhancement pattern can be nodular, linear, or curvilinear, as well as focal or diffuse [65] (Figure 3). New classification guidelines suggest that in addition to CSF cytology (positive; type 1, negative; type 2), categorizing LM imaging patterns as linear (subtype A), nodular (subtype B), both (subtype C), or neither (no imaging evidence of LM with a possible exception of hydrocephalus; subtype D) has prognostic significance [50,51]. To this effect, patients with type 2A or 2C nodular LM lesions have been reported to exhibit a worse prognosis compared to patients with non-nodular disease [50]. However, it is well documented that nodular LM lesions are associated with a surgical resection of parenchymal brain metastases treated with adjuvant stereotactic radiosurgery and have been shown in other studies to confer improved prognosis when compared to diffuse LM [49,66,67,68]. Further efforts to better define the prognostic and predictive relevance of distinct LM imaging patterns is an active area of study that can lead to a refined clinical and biological understanding of distinct LM entities. Importantly, LM can also affect different parts of the CNS, including the meninges surrounding the brain, within the ventricles, surrounding cranial nerves, or lining the spinal cord and its nerve roots [65]. The location of LM is thought to play a role in prognosis, with patients possessing cranial-only involvement exhibiting better outcomes than those with brain and spinal LM [69].

## 5. Treatment Approaches for LM

Identifying treatments that effectively manage LM has been extremely challenging. In part, the difficulty is associated with challenges and controversies in diagnosing LM and measuring the treatment response [70]. Surgical resection is rarely an option for leptomeningeal lesions due to their widespread dissemination in the subarachnoid space. For this reason, systemic and radiotherapy approaches remain the standard of care for treating LM.

### 5.1. Systemic and Intrathecal Therapies

To date, a small number of clinical trials have been reported specifically studying systemic treatments for breast cancer, lung cancer, and melanoma LM [71,72,73,74,75,76,77,78,79,80,81,82,83,84,85,86,87,88,89,90,91], a small subset of which were randomized studies [71,72,73,75,89,90]. Several of these studies employ an intrathecal administration of chemotherapy, meaning that the agents are injected directly into the CSF. This is performed via lumbar injection or with an Ommaya reservoir, a portal that is inserted from the exterior of the skull directly into the lateral ventricle to allow drug administration. A recent trial compared intrathecal liposomal cytarabine plus systemic chemotherapy versus systematic chemotherapy alone for the newly diagnosed breast cancer LM. This study revealed a LM progression-free and overall survival advantage in the population treated according to the trial protocol with intrathecal liposomal cytarabine plus systemic chemotherapy [89]. This finding is an important proof-of-principle supporting the use of intrathecal chemotherapy for patients with breast cancer LM and underscores the value of ongoing investigations into the intrathecal administration of novel agents. To this effect, clinical trials are underway to administer immunotherapy intrathecally, with early reports suggesting that this approach is adequately safe to begin investigating efficacy compared to systemic treatments [91]. In addition, the intrathecal administration of trastuzumab has also been shown in multiple studies to be capable of eliciting responses in the treatment of HER2+ breast cancer LM [92,93,94].

A systemic treatment with small-molecule targeted agents also plays an important role in the management of LM. These therapies include the standard targeted agents used for a particular tumor type defined by specific molecular features, such as abemaciclib (ER+) and trastuzumab (HER2+) for breast cancer LM, EGFR inhibitors in EGFR mutant lung cancer LM, and BRAF + MEK inhibition in BRAF mutant melanoma LM [51,83]. Most recently, the third-generation mutant EGFR inhibitor, osimertinib, has shown substantial activity in controlling LM from the EGFR mutant lung cancer [78]. Its activity is largely due to the improved CNS penetration compared to first-generation EGFR inhibitors (gefitinib and erlotinib) [85]. Similarly, new ALK inhibitor drugs that are effective in patients with ALK-translocated NSCLC (ceritinib and alectinib) display an improved CNS penetration compared to crizotinib [95,96].

With recent advancements in immunotherapy, these modalities are now being assessed for their efficacy in treating LM. A recent phase II single-arm trial assessed systemic pembrolizumab in the treatment of LM, predominantly in breast cancer patients, revealed that this treatment approach comes with an acceptable toxicity and promising activity [77]. It remains to be seen whether an intravenous or intrathecal delivery is optimal for patients with LM. While monoclonal antibody agents may not be able to effectively cross the blood-brain barrier, the rationale for intravenous delivery of immunotherapy for the treatment of LM is that extracranial immune cells may be accessible by the drug and then be able to home to the subarachnoid space. This concept has been shown in pre-clinical models to be a mechanism by which immunotherapy can function for parenchymal brain metastases in the presence of extracranial lesions [97].

### 5.2. Radiation Therapies

Radiation therapy remains a standard approach for the treatment of LM despite the lack of reported randomized clinical trials that assess the efficacy of this treatment modality. Typically, stereotactic radiosurgery (SRS) can only be effectively used for nodular LM lesions and may play a role in restoring CSF flow in patients with obstructive lesions [98]. Whole brain radiotherapy (WBRT) remains a commonly used radiation approach for treating patients with extensive LM or co-existing brain metastases, but observational data suggests that this approach offers little survival advantage and may be considered more of a palliative treatment [99,100]. Craniospinal irradiation is rarely used due to severe toxicities [51]. Novel strategies such as intrathecal administration of radioisotopes is an emerging strategy that may lead to important advances for the treatment of LM in the near-term [51].

## 6. Applying Pre-Clinical Animal Models of LM to Study Its Underlying Biology

To date, pre-clinical research of leptomeningeal metastasis from solid tumors has provided a limited understanding of this clinical entity. These efforts have been severely hampered by the fact that leptomeningeal metastases are typically not amenable to biopsy or surgical resection, leading to the use of CSF-derived tumors cells as a proxy to study LM. Rapid autopsies have been suggested as a novel approach to procure large quantities of LM tissue for research purposes [101]. Furthermore, no animal models of spontaneous LM from the primary tumor site of an adult solid tumor type have been reported thus far, rendering it impossible to study the process of LM from beginning to end. However, a number of studies have created animal models whereby tumor cells reach the leptomeninges via intra-cisternal, intra-carotid, intra-cranial, or intra-cardiac injection. While these models do not recapitulate the entire metastatic process, important insights have been gained from such models.

The first pre-clinical study to establish an animal model of LM involved the intracisternal injection of B16-melanoma cells and intrathecal injection of treatments [102]. This was followed by the development of metastatic melanoma models that, when injected into the carotid artery, had specific tropism to colonize the leptomeninges and not the brain parenchyma [103,104]. These publications were the first to suggest a distinct biological etiology in tumors metastatic to the leptomeninges compared to the brain parenchyma.

EGFR-mutant lung cancer cells injected into murine CSF via the cisterna magna have been shown to be more sensitive to osimertinib when compared to earlier generation EGFR inhibitors [105], supporting clinical findings [106]. Interestingly, these models have also revealed differential resistance mechanisms to gefinitib, with subcutaneous tumors frequently developing the gatekeeper T790M mutation, while LM tumors developed resistance by MET copy number gain [107]. This suggests the possibility for distinct influences from the leptomeningeal microenvironment in guiding the tumor evolution that may be differentially targeted.

To date, the most thorough pre-clinical investigation of LM from solid tumors involved the establishment of organotropic cell lines from breast and lung cancer that form LM when injected into the left cardiac ventricle of mice [13]. The authors demonstrate that in these models, cancer cells reach the leptomeninges through the choroid plexus and rely upon complement 3a signaling to breach the blood-CSF barrier, allowing the entry of mitogenic factors into the CSF to promote tumor cell survival. This study was groundbreaking in that it clearly demonstrates that LM is biologically distinct from parenchymal brain metastases. However, it leaves open questions about the differences between anatomical routes cancer cells use to reach the leptomeninges, since the choroid plexus is believed to be a rare site of entry for LM in humans [40]. This work has been extended by functionally validating another gene overexpressed in LM-tropic cell lines, lipocalin-2 (LCN2) [13,38]. This study demonstrated, using scRNAseq from patient LM alongside mouse models, that limiting quantities of iron in the leptomeningeal space is efficiently collected by cancer cells via LCN2 expression activated by macrophages in the CSF. These LM-tropic cell lines have also revealed distinct floating and adherent phenotypical states [108]. Floating LM cells display decreased proliferation but are enriched in the TCA cycle and electron transport chain signatures, a degree of metabolic flexibility that may provide rationale for how LM cells adapt to the limiting glucose in the CSF [108]. In mouse models, the implantation of floating LM cells resulted in the shortened overall survival compared to adherent cells, mirroring the results observed in a retrospective patient dataset and the LM guidelines demonstrating that patients with cytology (+) MRI (-) LM exhibited inferior prognosis than those with MRI (+) disease [50,108]. This suggests a model whereby LM cells exist in equilibrium between floating within the CSF and adherent to the leptomeninges, with floating cells as the drivers of aggressive disease biology.

Other studies have demonstrated the development of LM secondary to parenchymal brain metastases using murine models. Palmieri et al. demonstrated that both parenchymal and leptomeningeal brain metastases are observed when brain-metastatic MDA-MB-231 cells are introduced via intracardiac injection [109]. The ability of MDA-MB-231 cells to form LM upon intracranial injection was confirmed by Allen et al., who observed leptomeningeal dissemination when MDA-MB-231, but not MDA-MB-468 cells, were implanted [44]. These findings were expanded upon using a cohort of patient-derived xenograft models of brain metastases injected intracranially, which show differential propensity to form LM based upon the invasion pattern in the brain metastatic lesion (minimally versus highly invasive) and expression of CIRBP [45]. This suggests that cellular processes and molecular mediators that are present in only a subset of cancers may be involved in leading established parenchymal brain metastases to subsequently colonize the leptomeninges. Furthermore, the invasion status in surgically resected brain metastases may represent a biomarker to predict the subsequent development of LM.

Together, a number of unique model systems exist to study LM, that include intracardiac, intracranial, and intracisternal injection of cancer cells. While each of these routes of injections comes with their individual caveats, they have together provided significant insight into the biological mechanisms by which LM reach the leptomeninges, survive in the subarachnoid space, and evolve in the presence of the treatment. Future work must aim to develop improved models of this entity that stem from primary tumors that recapitulate the entire metastatic process, including patient-derived xenografts, transgenic and syngeneic models.

## 7. Avenues of Future Investigation and Conclusions

LM is becoming increasingly appreciated by the clinical and research communities as an important area of need in clinical and fundamental cancer research, as evidenced by the rapidly rising number of publications in this area. However, many knowledge gaps exist that can be explored for the benefit of patients with LM.

The diagnosis of LM remains primitive. While the combination of neurological evaluation, MRI, and CSF cytology are able to diagnose LM, emerging technologies employing liquid biopsy approaches are poised to make important advances in this area. A future study of different patterns of LM observed by MRI are warranted to refine the existing classification system for LM informed by the disease biology and differential patient outcomes.

The route of entry by which cancer cells reach the subarachnoid space has yet to be extensively studied. Cancer cells employing the recently discovered glymphatic system or perivascular spaces to reach the subarachnoid space remain a distinct possibility. A study of these features requires a combination of clinical specimens, multiple robust pre-clinical animal models that recapitulate the LM process from beginning to end, and advanced imaging technologies to visualize these small anatomical structures with high resolution. In addition, the potential shuttling of cancer cells between the subarachnoid space and brain parenchyma remains a tantalizing explanation for resistance to intrathecal and systemic treatments.

Systemic therapies for LM remain cancer-type specific. Pre-clinical studies have suggested the possibility of exploiting specific features of LM and its microenvironment, with antagonists of the complement 3a receptor or iron chelators in development [13,38]. The translation of these studies towards clinical trials may lead to the first targeted treatments specifically for patients with LM.

Together, there are many critical questions remaining in the study of LM. With rapidly expanding efforts to study this entity both clinically and fundamentally, it is expected that rapid progress will occur in the short order. It is essential that these findings are translated to the patient care, given the devastating impact of LM on patients with advanced solid cancers.

## 8. Conclusions

Leptomeningeal brain metastases represent clinically and biologically distinct disease entities when compared to more highly prevalent parenchymal brain metastases. Challenges remain with respect to improved diagnosis and treatment of leptomeningeal metastases. Exciting opportunities also exist to better understand the underlying biology of this disease and apply this knowledge to the generation of therapeutic strategies targeting leptomeningeal metastases.

## Figures and Tables

**Figure 1 cancers-13-00732-f001:**
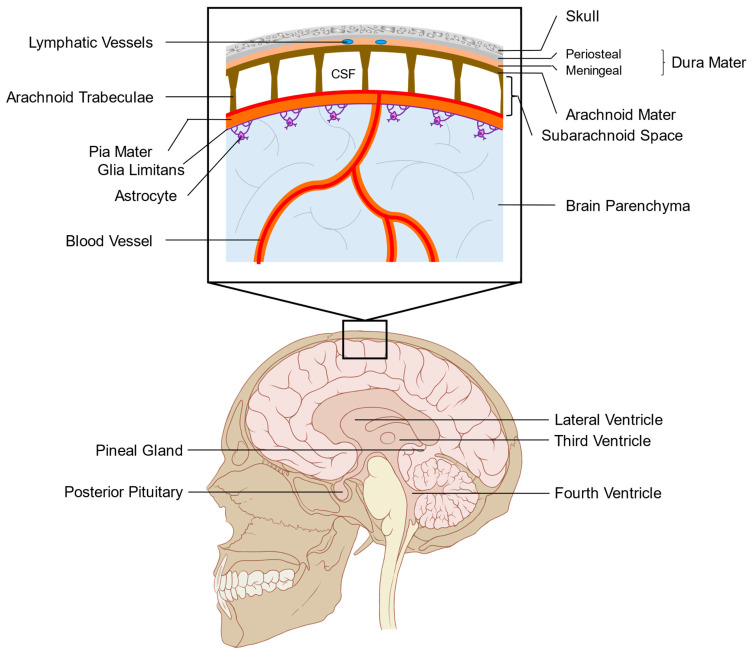
Anatomy of the leptomeninges and surrounding structures. Leptomeningeal-metastatic cancer cells exist in the subarachnoid space, between the arachnoid and pia mater, deep in the dura mater and skull. Cancer cells may be adherent to the walls of the subarachnoid structures or survive in suspension in the cerebrospinal fluid (CSF). CSF is created in the choroid plexus of the lateral and fourth ventricles and flows through the ventricular system into the subarachnoid space. The subarachnoid space is separated from the brain parenchyma by the glia limitans consisting of astrocytic foot processes. The leptomeninges surrounds the entire brain and spinal cord, with notable exceptions including circumventricular organs such as the pineal gland and posterior pituitary.

**Figure 2 cancers-13-00732-f002:**
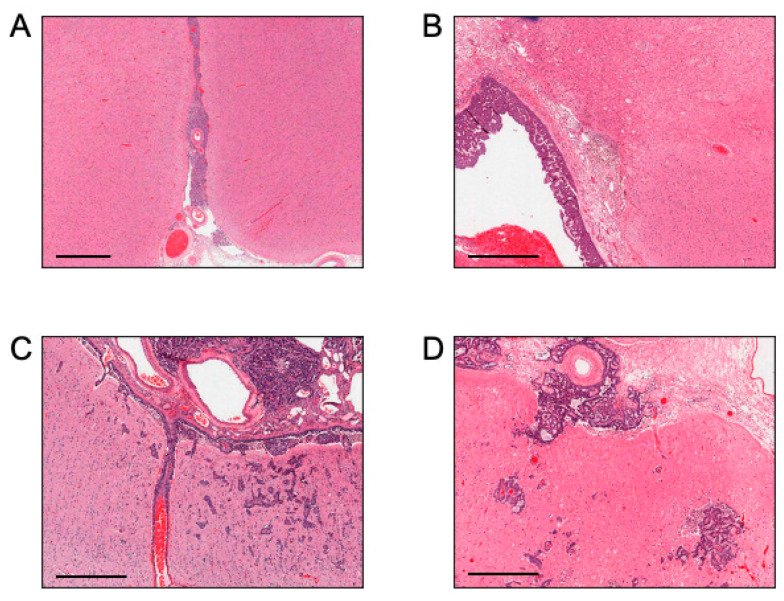
Different appearances of leptomeningeal metastasis (LM) by histopathology in lung cancer patients. H&E stains demonstrating cancer cells (**A**,**B**) exclusively in the subarachnoid space and (**C**,**D**) in the subarachnoid space with communication of cancer cells with the brain parenchyma through local invasion. Scale bars: 1 mm.

**Figure 3 cancers-13-00732-f003:**
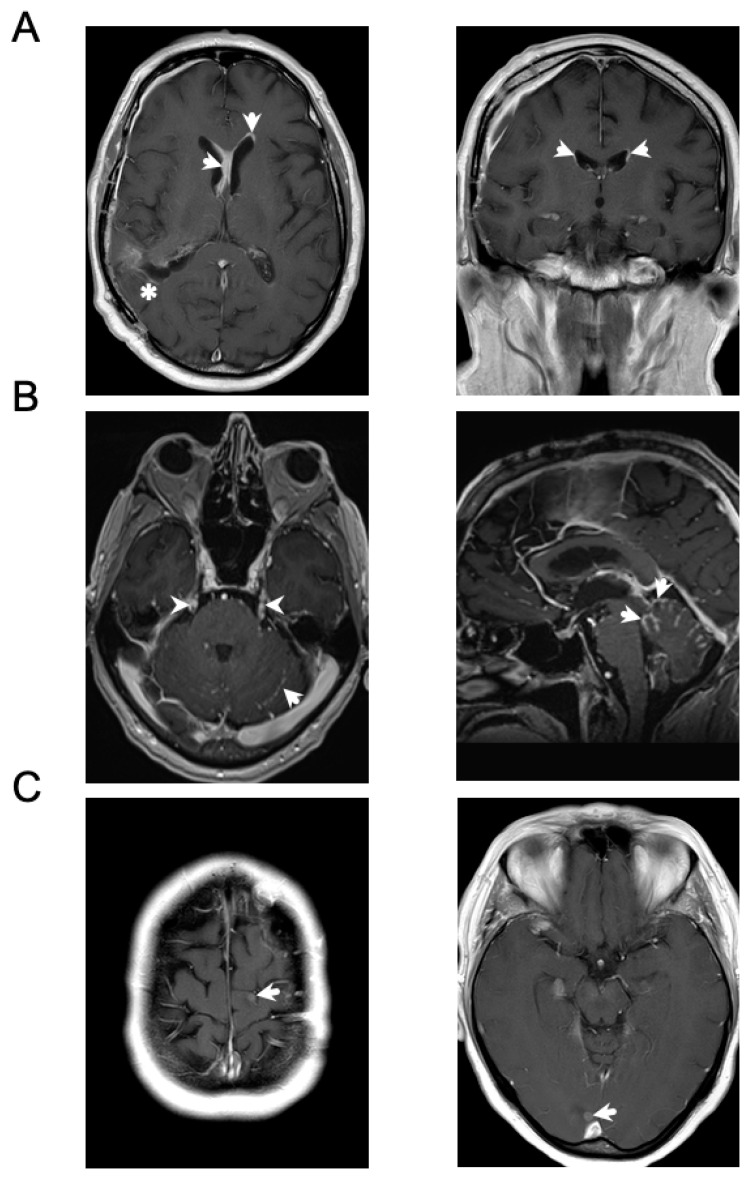
Different appearances of LM by magnetic resonance imaging (MRI). (**A**) Patient with melanoma brain metastases. Axial (left) and coronal (right) T1 post contrast images demonstrate ependymal metastases along the lateral ventricles (arrows). There are also post-surgical changes related to the prior resection of a parietal parenchymal metastasis (asterisk). (**B**) Patient with lung cancer brain metastases. Axial (left) and sagittal (right) T1 post contrast images demonstrate linear, “classical”, leptomeningeal metastases along the trigeminal nerves (arrowheads) and “sugarcoating” the cerebellum (arrows). (**C**) Patient with lung cancer brain metastases. Axial T1 post contrast images demonstrate multiple small, superficially-located lesions, representing nodular leptomeningeal metastases

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
