# Peer review of "The Underlying Biology and Therapeutic Vulnerabilities of Leptomeningeal Metastases in Adult Solid Cancers"

_cancers, 2021, doi:10.3390/cancers13040732_

Round 1

Reviewer 1 Report

Dear authors,

thank you very much for this overview of the current knowlegde concerning the leptomeningeal metastases (LM).

Your manuscripts includes a scientifical/molecular data to the topic and important issues of the clinical Management strategies. Furthermore it presents an overview of the potential future management options of the LM.

I recommend you to optimize following issues:

  • please optimize the structure of the description of the "systemic and intrathecal therapies" of LM according to the primary Tumor (lung cancer, breast cancer, melanom)
  • please give an overview of the role of Methotrexate as a intrathecal substance

Greetings,

Reviewer 2 Report

The authors report on the underlying biology and therapeutic vulnerabilities of leptomeningeal metastasis (LM) in adult solid cancers.

This is a very well written review on a topic most likely many clinicians are not aware of.

Comments

Section Introduction - the authors mention that surgical resection of parenchymal brain metastases is a risk factor associated with development of LM. Is this simply due to spillage of tumour cells or is there a more sophisticated reason for the association?

Do patients with LM have other (non-brain) distant metastases, or are there also cases of LM in an oligometastatic setting?

Is there any sex and/or age dependency?

Figure 2 – which magnification was used?

Section Diagnosis – has 18F FDG PET/CT any role in diagnosis of LM?

Avenues of future investigations – Has any research been done on genomics of LM? If so, are there any differences between LM and parenchymal brain metastases?
